

# Ginsenoside Rb1 enhanced immunity and altered the gut microflora in mice immunized by H1N1 influenza vaccine

Chuanqi Wan[1], Rufeng Lu[2], Chen Zhu[3], Haibo Wu[4], Guannan Shen[2], Yang Yang[2], Xiaowei Wu[2], Bangjiang Fang[1,5] and Yuzhou He[2]

[1] Department of Emergency, Longhua Hospital, Shanghai University of Traditional Chinese Medicine, Shanghai, Xuhui, China
[2] Department of Emergency, The First Affiliated Hospital of Zhejiang Chinese Medical University, Hangzhou, Shangcheng, China
[3] Department of ICU, The First Affiliated Hospital of Zhejiang Chinese Medical University, Hangzhou, Shangcheng, China
[4] The First Affiliated Hospital, Zhejiang University, State Key Laboratory for Diagnosis and Treatment of Infectious Diseases, Collaborative Innovation Center for Diagnosis and Treatment of Infectious Diseases, School of Medicine, National Clinical Research Center for Infectious Diseases, Hangzhou, Shangcheng, China
[5] Institute of Critical Care, Shanghai University of Traditional Chinese Medicine, Shanghai, Xuhui, China

Corresponding authors
Bangjiang Fang, fangbji@163.com
Yuzhou He, hyzxy1995@126.com

## ABSTRACT

**Background**. Influenza is an acute infectious respiratory disease caused by the influenza virus that seriously damages human health, and the essential way to prevent influenza is the influenza vaccine. Vaccines without adjuvants produce insufficient specific antibodies and therefore require adjuvants to boost antibody titers. Microbes and hosts are a community that needs to "promote bacteria," which could provide new value for the immune effect.

**Methods**. (1) The H1N1 influenza vaccine, in combination with Ginsenoside Rb1, was co-injected into mice intraperitoneally (I.P.). Then, immunoglobulin G and antibody subtype levels were tested by enzyme-linked immunosorbent assay (ELISA). Moreover, mice were infected with a lethal dose of the H1N1 influenza virus (A/Michigan/45/2015), and survival status was recorded for 14 days. Lung tissues were stained by hematoxylin and eosin (H&E), and ELISA detected inflammatory factor expression levels. (2) Mice were immunized with Ginsenoside Rb1 combined with quadrivalent influenza inactivated vaccine(IIV4), and then IgG levels were measured by ELISA. (3) Fresh stool was collected for fecal 16S rDNA analysis.

**Results**. Ginsenoside Rb1 boosted IgG and antibody subtypes in the H1N1 influenza vaccine, improved survival of mice after virus challenge, attenuated lung histopathological damage, and reduced inflammatory cytokines expression in IL-6 and TNF-$\alpha$. The results of 16S rDNA showed that Rb1 decreased species diversity but increased species richness compared to the PBS group and increased the abundance of *Akkermansiaceae* and *Murbaculaceae* at the Family and Genus levels compared with the HA+Alum group.

**Conclusion**. Ginsenoside Rb1 has a boosting effect on the immune efficacy of the H1N1 influenza vaccine and is promising as a novel adjuvant to regulate the microecological balance and achieve an anti-infective effect.

## INTRODUCTION

The high incidence of influenza poses an enormous burden to the economy and society. Research on influenza vaccines is highlighted by their ability to boost the body's immune antibody levels (*Keilman, 2019*; *Webster & Govorkova, 2014*). Adjuvants can enhance the body's immune response and boost the body's immune antibody titer, resulting in adequate immune protection (*Wang, Liu & Zhao, 2021*). Therefore, developing safe and effective vaccine adjuvants is a research hotspot in infectious diseases (*Sun et al., 2018a*).

Ginsenoside Rb1 ($C_{54}H_{92}O_{23}$, Rb1), extracted from the roots of ginseng, was the main active ingredient. Current studies of Rb1 in immune effects show it has antioxidant, anti-inflammatory, and immune-enhancing properties (*An et al., 2021*; *Lu et al., 2020*). It has an excellent adjuvant effect on immune response and can elevate humoral and cellular immune effects (*Rivera et al., 2005*; *You et al., 2022*). Studies have shown that Rb1 may act as a potential prebiotic agent to regulate specific intestinal microorganisms and play a role in diseases, such as protecting against diabetes-associated metabolic disorders (*Li et al., 2018*; *Zhou et al., 2023*), suppressing colon cancer (*Wang et al., 2023*; *Yao et al., 2018*), improving intestinal aging (*Lei et al., 2022*), exerting neuroprotective effects(*Chen et al., 2020*; *Zeng et al., 2018*), improving metabolic disorder in high-fat diet-induced obese mice (*Bai et al., 2021*; *Yang et al., 2021*; *Zou et al., 2022*)and hyperlipidemia (*Bai et al., 2021*; *Lianqun et al., 2021*). Moreover, prebiotics can selectively promote the proliferation of bacterial strains, thereby improving the biological conversion rate and bioavailability of ginsenosides in the body (*Shen et al., 2018*; *Zhang et al., 2021*).

Immune responses to vaccines vary among individuals. Recently, one factor associated with this variability in vaccine responses is microbiota composition, including the gut and lung (*Chen et al., 2021*; *Oh & Seo, 2023*; *Seong et al., 2023*). Studies have reported relationships between better vaccine responses and specific bacterial taxa in humans, so prebiotics and probiotics as vaccine adjuvants may improve respiratory virus vaccine responses (*Zhuang et al., 2023*). When the permeability of the intestinal barrier is disrupted, bacterial fragments and metabolites will enter the circulation and reach distant organs, which can affect immune cells, such as microglia and thymocytes (*Damiani, Cornuti & Tognini, 2023*). Therefore, paying more attention to the gut microbiota may be one way to increase vaccine responses against respiratory pathogens (*Hong, 2023*).

In summary, we hypothesize that Rb1 has an immune-enhancing effect, correcting the species richness and diversity of gut microbiota in influenza vaccines. Our research objective is to investigate the role of Rb1 as an adjuvant for the H1N1 influenza vaccine and IIV4 in enhancing immunity and reducing mortality, and to investigate whether the combination of the H1N1 influenza vaccine and Rb1 can alter the gut microbiota, ultimately filling the research gap in this area. Therefore, we will explore the following two aspects. Firstly, we investigate the immune-boosting effect of ginsenoside Rb1 in influenza vaccines, including antibody titers, survival rate changes, lung histopathological changes, and serum inflammatory factor expression levels. Secondly, we performed a screen for gut microbial diversity, richness changes, and gut-dominant microbes in mice after the second immunization.

## MATERIALS & METHODS

### Animal experiments

A total of 96 BALB/c mice (male, 6–8 weeks, 20 ± 4 g) were obtained from the Shanghai SLAC Laboratory Animal Co., Ltd. All mice were housed in a sterile environment with controlled temperature and humidity with a 12 h light/dark cycle and were allowed free access to food and water. In terms of mouse grouping, these mice were randomly divided into the following groups. Group I: PBS (control group, $n = 7$), Group II: HA-3 μg (antigen group, $n = 7$), Group III: HA+Alum-100 μg ($n = 7$), Group IV: HA+Rb1-70 μg ($n = 7$), Group V: HA+Rb1-200 μg ($n = 7$), Group VI: HA+Rb1-600 μg ($n = 7$). The above six groups (Section A) were used for survival changes observation, and the same six groups (section B, $n = 6$) were used for collecting lung tissue or serum. The purpose of grouping in this way is to reduce interference caused by overlapping experimental operations. Moreover, we set up three groups of mice ($n = 6$) to evaluate the antibody-boosting effect of Rb1-200 μg in IIV4. The Animal Ethics Committee of Zhejiang Chinese Medical University reviewed and approved the animal study (IACUC-20220725-37). Finally, the mice were euthanized due to cervical dislocation.

### Chemicals and antibodies

Ginsenoside Rb1 was purchased from Beijing Solarbio Technology Co., Ltd. (Beijing, China) with product number SA9790 and purity ≥ 98%. Influenza A virus antigen and IIV4 were purchased from Zhejiang Tianyuan Biopharmaceutical Co., Ltd; the A/Michigan/45/2015 virus was stored in our laboratory. Aluminum hydroxide adjuvant was purchased from Thermo Fisher Scientific Co., Ltd (Waltham, MA, USA). Secondary antibodies were purchased from Southern Biotech. The cytokine kit was purchased from Beijing 4A Biotech Co., Ltd.

### Mouse immunization and challenge

The H1N1 vaccine (3 μg, A/Michigan/45/2015) was combined with different doses of adjuvant using PBS configured to 200 uL, and mice were injected (I.P.) on days one and 14. Then, 200 uL of whole blood was collected from the inner canthal vein of group B mice on day 28, centrifuged for 10 min, and the separated serum was used for subsequent ELISA, HI, and MN assays. Next, the influenza A virus ($5 \times MLD50$, A/Michigan/45/2015) infects all mice by nasal instillation on day 35 and collects whole blood and lung tissue from section B mice on day 41 (day six post-infection). Section A mice were observed for body weight loss and survival rate for 14 days. When the mouse's body weight was less than 30% of its initial weight, we used the method of cervical dislocation to euthanize it.

### Serum IgG, antibody subtype titers, and inflammatory cytokines by ELISA

ELISA was performed as previously described (*Wu et al., 2018*). First, the coating antigens were A/Michigan/45/2015, A/Texas/50/2012, B/Brisbane/60/2008, and B/Phuket/3073/2013 (B/Yamagata), respectively. The antigen was diluted to 0.2 μg/mL with coating buffer, and then blocking buffer (Biopanda, BB-001) was added to the

96-well microplate plate. Mouse sera were diluted $2^n$ with PBS (1,000, 2,000, 4,000, 8,000, 16,000, 32,000, *etc.*). Peroxidase labeled Goat anti-mouse secondary antibody IgG (1031-05; SouthernBiotech) was added, followed by incubation at 37 °C for 30 min, then TMB (TMB-S-004; Biopanda) was added to a 96-well microplate plate, and finally, stop solution (Biopanda, STP-001; Biopanda) was added. OD values were determined at 450 nm and interpreted as positive with >2.1-fold negative control values. For serum antibody subtypes, the corresponding secondary antibodies were replaced with IgG1a (1071-05; SouthernBiotech), IgG2b (1091-05; SouthernBiotech), and IgG3 (1101-05; SouthernBiotech), and the test methods were the same as described above. As for inflammatory cytokines, we measured the expression levels of IL-6 (CME0006; 4A Biotech) and TNF-α (CME0004; 4A Biotech) by ELISA in mice according to the protocol.

## Histopathological examination

For lung histopathology, mouse lung tissues were harvested, fixed in 10 ml 10% formalin, embedded in paraffin, and sliced into four μm-thick sections with a slicer. The sections were stained with hematoxylin and eosin. Finally, histological observations were performed by microscopy.

## Microbial diversity analysis
### High-Throughput sequencing and data quality control

The fresh fecal samples were collected from mice in PBS, HA, HA+Alum, and HA+Rb1-200 μg groups. The V3–V4 hypervariable region of the bacteria 16s rRNA was amplified as a template, which the forward primer 515F (5′-GTGCCAGCMGCCGCGGTAA-3′) and the reverse primer 806R (5′-GGACTACHVGGGTWTCTAAT-3′) (*Caporaso et al., 2011*). The PCR components and reaction process as previously described (*Wan et al., 2021*), briefly described, 50 μL mixture includes Phusion High Fidelity PCR main mixture (25 μL), DNA template (10 μL), forward primer (3 μL), reverse primer (3 μL), DMSO (3 μL), ddH$_2$O (6 μL). And, the cycle is briefly described as initial denaturation at 98 °C for 30 s, followed by 98 °C for 15 s, 58 °C for 15 s, 72 °C for 15 s, and finally at 72 °C for 1 min. Each was independently amplified three times. Then, the amplification products were then extracted using gel electrophoresis and further purified using the Agencourt AMPure XP Beads (Beckman Coulter, Indianapolis, IN, USA) and quantified using the PicoGreen dsDNA Assay Kit (Invitrogen, Carlsbad, CA, USA) according to the protocol. Next, purified amplicons were pooled in equimolar and sequenced on an Illumina NovaSeq6000 platform for generating $2 \times 150$ bp paired-end reads. Filtering the raw data could improve the subsequent bioinformatics analysis accuracy. As shown in Table 1, raw tags were filtered out in low-quality data, adaptor, or PCR errors *via* Vsearch (v2.15.0). After removing duplicates, singleton indels, and chimeras (*Haas et al., 2011*), tags matched ASVs (Amplicon Sequence Variant) were generated that could be used for subsequent analysis. ASVs table was obtained by the UNOISE2 method *via* Vsearch (v2.4.4).

### Diversity and richness analysis of mice gut microbiota

In this study, alpha diversity reflects the abundance and diversity of the gut microbiota in each mouse, including Chao1, ACE, Shannon, and Simpson, by software Quantitative

**Table 1  Statistical results of sequencing.**

| Group | Raw_tags | Singleton | Tags matched ASVs | ASVs |
|---|---|---|---|---|
| PBS | 121156 ± 8409 | 99 ± 44 | 118851 ± 7834 | 679 ± 163 |
| HA | 124701 ± 69 | 134 ± 8 | 121599 ± 977 | 781 ± 89 |
| Alum+HA | 124701 ± 167 | 98 ± 2 | 119973 ± 1441 | 771 ± 57 |
| Rb1+HA | 124669 ± 56 | 126 ± 11 | 120302 ± 3081 | 813 ± 99 |

**Notes.**

Raw_tags, raw sequence data per group; Singleton, the number of single sequences without a match; Tags matched ASVs, effective sequence data; ASVs, number of ASVs per group.

Insights Into Microbial Ecology (QIIME.2) (*Caporaso et al., 2010*). Beta diversity analysis reflected differences in microbial communities in each group. Besides, beta diversity mainly includes the Venn diagram, Principal Component Analysis (PCA), and Bray-Cruits dissimilarity matrix. However, we just used the above ways to distinguish whether there is a significant difference in the intestinal flora of mice between groups, which is not enough. We also rely on the Linear discriminant analysis Effect Size (LEFSe; |LDA| > 2) to distinguish the specific difference flora between groups.

## Statistical analysis

Statistical analysis was conducted using SPSS 22.0 (SPSS Inc., Chicago, IL, USA) and GraphPad Prism 8 (GraphPad, San Diego, CA, USA), and the measurement data are expressed as mean ± SEM (Standard Error of Mean). The student's $t$-test is used to compare two different groups. Analysis of variance (ANOVA) to compare three or more groups. In addition, the Kruskal–Wallis rank sum test was used in alpha diversity. The significance of differentiation of microbiota structure among groups was assessed by PERMANOVA (Permutational multivariate analysis of variance), non-parametric Kruskal–Wallis rank sum test, Wilcoxon rank sum test, and linear discriminant analysis (LDA) which was used in Beta diversity. Differences were considered statistically significant at the $p$-value < 0.05.

## RESULTS

### Immune effects of Ginsenoside Rb1 in combination with H1N1 influenza split vaccine

This study used the H1N1 influenza vaccine combined with three doses (70 µg, 200 µg, and 600 µg) of ginsenoside Rb1 or aluminum adjuvant (100 µg) to immunize BALB/c mice. Then, to investigate the immune boosting effect of ginsenoside Rb1 on the vaccine, observe mice's body weight and survival rate changes after viral infection, as shown in Fig. 1. The vaccination and blood collection schedule is shown in Fig. 1A. As shown in Fig. 1B, we found that the IgG antibody enhancement effect of the HA group was significantly higher than that of the PBS group. Moreover, the levels of IgG antibodies in mice increased when Rb1 was added, and the immune enhancement effect of Rb1 showed a dose–effect relationship, but the difference was insignificant. Then, we explored the antibodies IgG1, IgG2b, and IgG3 (Figs. 1C–1E), which showed that Rb1-600 µg had a significant antibody-enhancing effect compared to the HA group ($P < 0.01$). At day 35, mice were infected using a lethal dose of influenza virus and then observed for survival. As shown in Figs. 1F–1G, the results

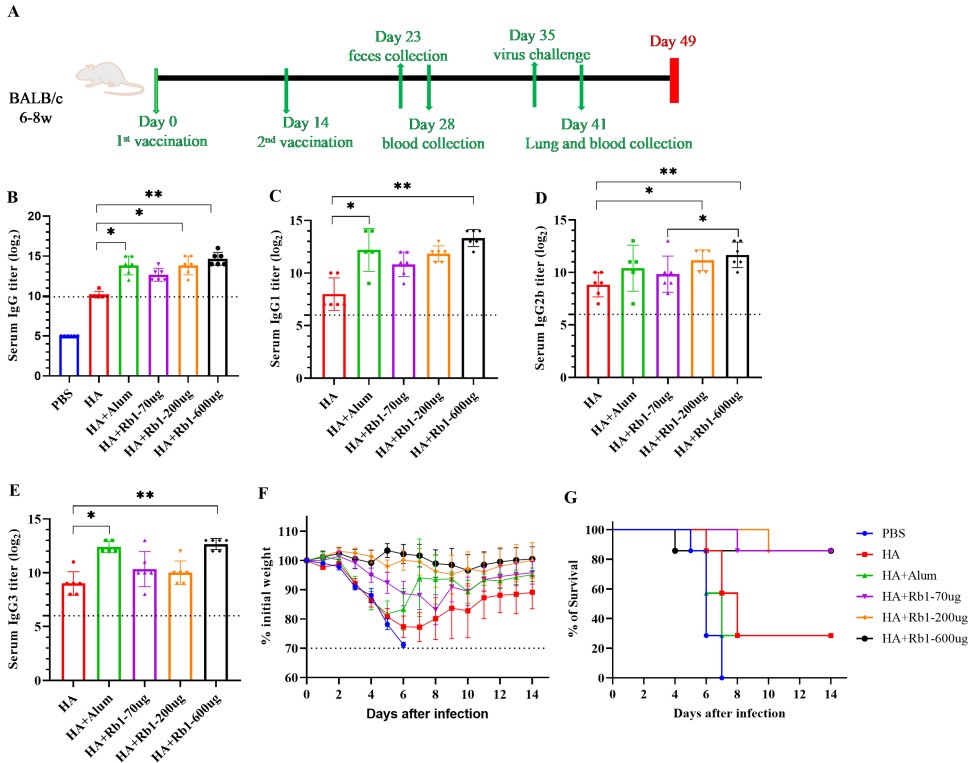

**Figure 1** **Adjuvant effects of Ginsenoside Rb1 or Alum in combination with HA antigen.** PBS or 3 μg of antigens with or without adjuvant (Alum-100 μg, Rb1-70 μg, Rb1-200 μg, Rb1-600 μg) were injected into mice *via* I.P. route twice every two weeks. The vaccinated mice were challenged with 5×MLD50 of the virus (H1N1, A/Michigan/45/2015). (A) Vaccination, blood collection, feces collection, and challenge schedule. (B) IgG antibody response to H1N1 antigen. (C–E) IgG antibody subtype response to H1N1 antigen. The endpoint dilution of each sample was expressed as the absorbance value close to 2.1 times that of the negative control group. (F–G) The weight changes and survival rates against the virus. The dashed line means losing 30% weight loss. (* $P < 0.05$, ** $P < 0.01$).

showed that the mice in the PBS group had apparent weight loss and all sacrificed on the seventh day, but the mice in the HA group weight gradually recovered, and the mice in the Rb1 group had a higher survival rate (85.7%, six mice survived/one mouse died) than those in the HA group (28.6%, 2 mice survived/5 mice died). Moreover, we found that mice in the Alum group also had a high mortality rate (28.6%), highlighting the predominance of mice in the RB1 group (85.7%).

## Effects of ginsenoside Rb1 on Lung pathology and inflammatory cytokines in post-infection mice

Mouse lung tissue pathology directly reflects the protective capacity of the vaccine against the influenza virus. As shown in Fig. 2A, the mice in the model group had severe congestion and damage in alveolar morphology. Mice in the HA group showed slight congestion compared to mice in the model group, but alveolar damage did not improve (Fig. 2B). Mice in the HA+Alum group had improved alveolar morphology and less congestion than those in the HA group (Fig. 2C). Importantly, mice in the HA+Rb1 group showed the
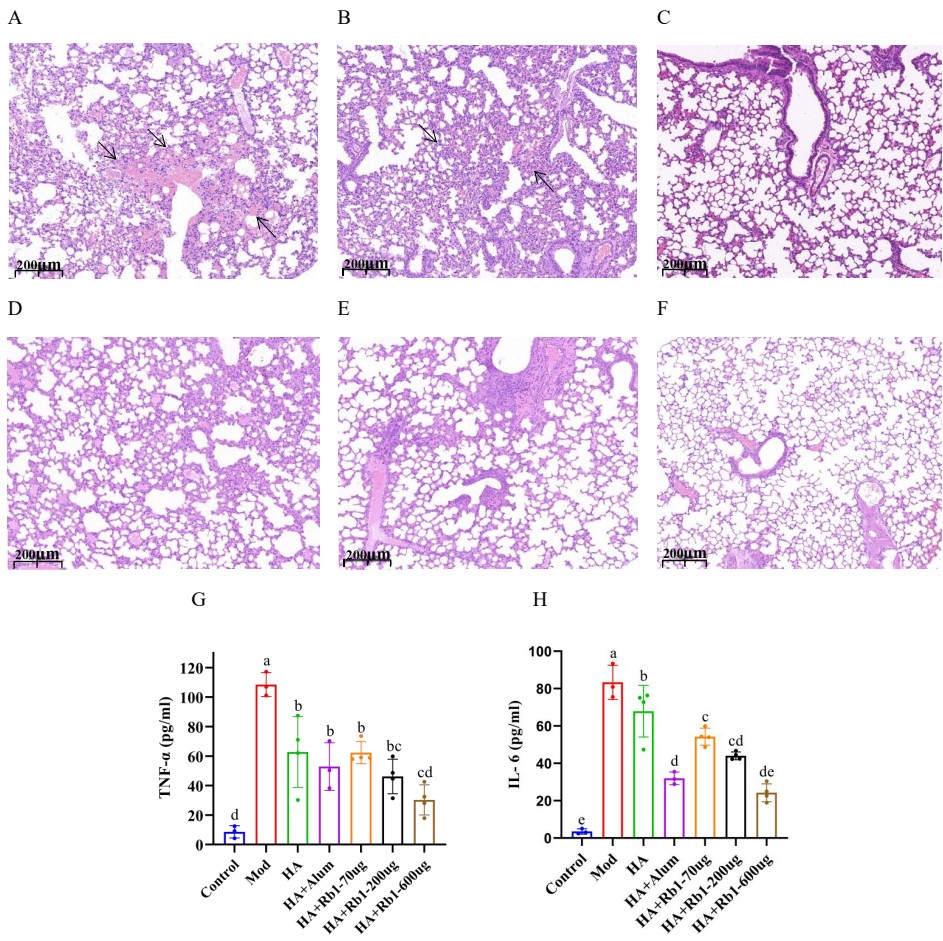

**Figure 2 Ginsenoside Rb1 attenuates severe lung injury and TNF-α and IL-6 levels during influenza virus infection.** Images are lung histopathology at day 6 of infection, 10× magnification (scale bar = 200 μm). (A) PBS (the model group). (B) H1N1 influenza vaccine-3ug(HA group). (C) H1N1 influenza vaccine+Alum adjuvant(HA+Alum group). (D) H1N1 influenza vaccine+Rb1-70 μg (HA+Rb1-70 ug). (E) H1N1 influenza vaccine+Rb1-200 μg (HA+Rb1-200ug). (F) H1N1 influenza vaccine+Rb1-600 μg (HA+Rb1-200ug). (G, H) Cytokine expression of TNF-α and IL-6 were measured in serum from mice six days after virus infection, n = 3–4. (*P < 0.05).

same trend as the Alum group (Figs. 2D–2F). We collected whole blood from mice before collecting lung tissue, which was used to measure levels of the inflammatory cytokines TNF-α and IL-6 (Figs. 2G–2H). The results showed that the inflammatory cytokines after vaccine immunization decreased significantly compared with those in the model group (P < 0.05). The adjuvant group had a better effect in reducing TNF-α and IL-6 expression, which provided a basis for survival protection after infection in mice compared with the HA group.

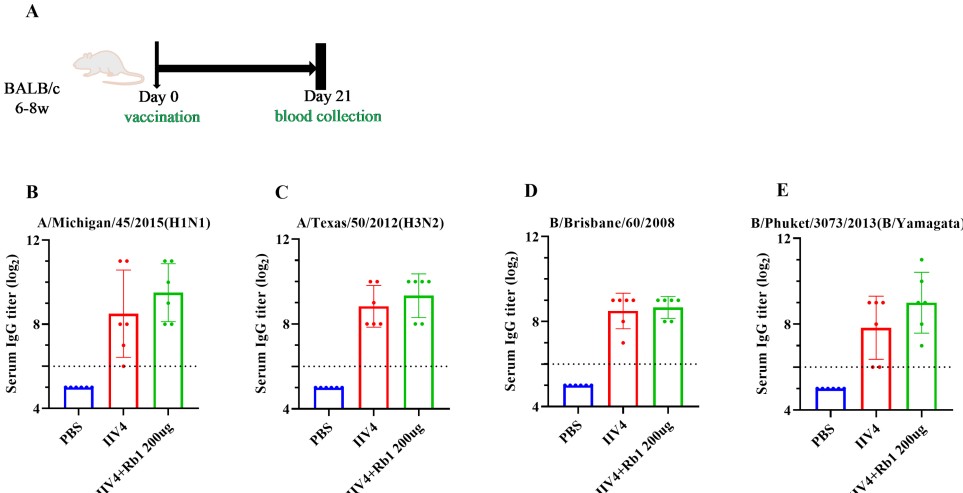

**Figure 3 Adjuvant effect of Ginsenoside Rb1 on IIV4.** (A) Vaccination, blood collection. PBS or 6ug of IIV4 (1.5 ug of each antigen derived from four strains) with or without Rb1 were once injected into mice *via* the I.P. route. Sera were collected three weeks after the vaccination. (B–E) Serum IgG titers for four different influenza strains ($n = 6$).

## Immune effects of ginsenoside Rb1 in combination with inactivated influenza virus quadrivalent vaccine

Mice were immunized with ginsenoside Rb1 in combination with IIV4, following the scheme in Fig. 3A, to investigate the effects on mouse IgG boosting. The results showed that mice in the Rb1 group had better immune boosting in A/Michigan/ 45/2015 (HIN1) and B/Phuket/3073/2013 (B/Yamagata) and limited in A/Texas/50/2012 (H3N2) and B/Brisbane/60/2008, compared with IIV4 alone (Figs. 3B–3E).

## Results of fecal 16s rDNA analysis
### Results of data control
A total of 2,971,370 raw reads were obtained from 24 stool samples of mice from the Illumina NovaSeq6000 sequencing platform. The mean number exceeded 120,000 in each group, as shown in Table 1. Singleton reflects the percentage of raw reads removed because of low quality. Ultimately, effective tags are harvested for subsequent analysis.

### Ginsenoside Rb1 induced diversity and richness variations in gut microbiota
Shannon and Simpson reflect the diversity of the microbiota, Chao1, and ACE reflect the richness of the microbiota. We found that Rb1 intervention decreased species diversity (Figs. 4A–4B) but increased species richness (Figs. 4C–4D) compared to the PBS group. β-diversity analysis reflected the similarities and differences of the gut microbiome in each mouse. Thus, the PCoA results based on the Bray Curtis distance showed that the variances of the samples among the four groups were statistically significant ($P < 0.01$).

### Gut microbiota variations in response to ginsenoside Rb1 intervention
We have explored the diversity and richness of samples from the four groups, but this was not enough, and we also discuss differences in the microbiota abundance on the

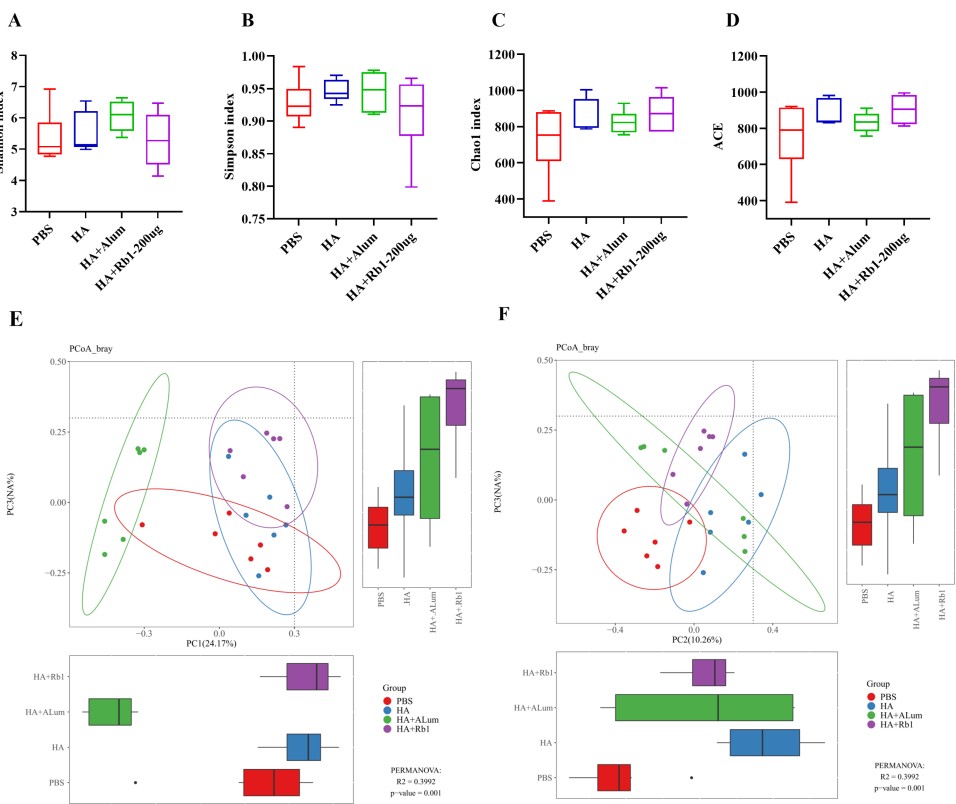

**Figure 4** **Ginsenoside Rb1 altered fecal microbial diversity in immunized mice.** 16S rRNA sequencing of the fecal samples in control (PBS), the model (HA), the Alum administration (HA+Alum), and the Rb1 administration groups (HA+Rb1) were detected at 23 dpi ($n = 6$). Alpha-Diversity was evaluated based on the Shannon (A), Simpson (B), Chao1 (C), and ACE (D) indices of the operational taxonomic unit (OTU) levels. Beta-Diversity was evaluated based on the principal coordinate analysis (PCoA) (E, F).

Family level (Fig. 5A) and the Genus level (Fig. 5B). We found that the abundance of *Akkermansiaceae* increased after the Rb1 intervention at the Family and Genus levels (compared to the other three groups). The abundance of *Murbaculaceae* in the HA+Alum group decreased at the Family and Genus levels (compared to the HA group), but this can be reversed after Rb1 intervention. As seen in Figs. 5C, and 6, the dominant flora of the Rb1 group included *Desulfovibrionaceae*, *Desulfovibrionales*, *Desulfovibrionia*, *Akkermansiaceae*, Verrucomicrobiales, Verrucomicrobiae.

## DISCUSSION

Vaccination is one of the most effective ways to control the spread of infectious diseases, extending the lifespan of animals (*Lambe, 2012*; *Schultz-Cherry & Jones, 2010*). The titers of specific antibodies generated by antigens may not be sufficient, resulting in poor defense in response to viral infection, so adjuvant studies are urgently needed (*Fan et al., 2018*; *Wu et al., 2016*). Polysaccharides extracted from traditional Chinese medicinal plants with biocompatible, degradable, and safe regulatory properties (*Gao & Guo, 2023*; *Wu et al., 2017*; *Zhao et al., 2023*). Therefore, it is essential to study Traditional Chinese

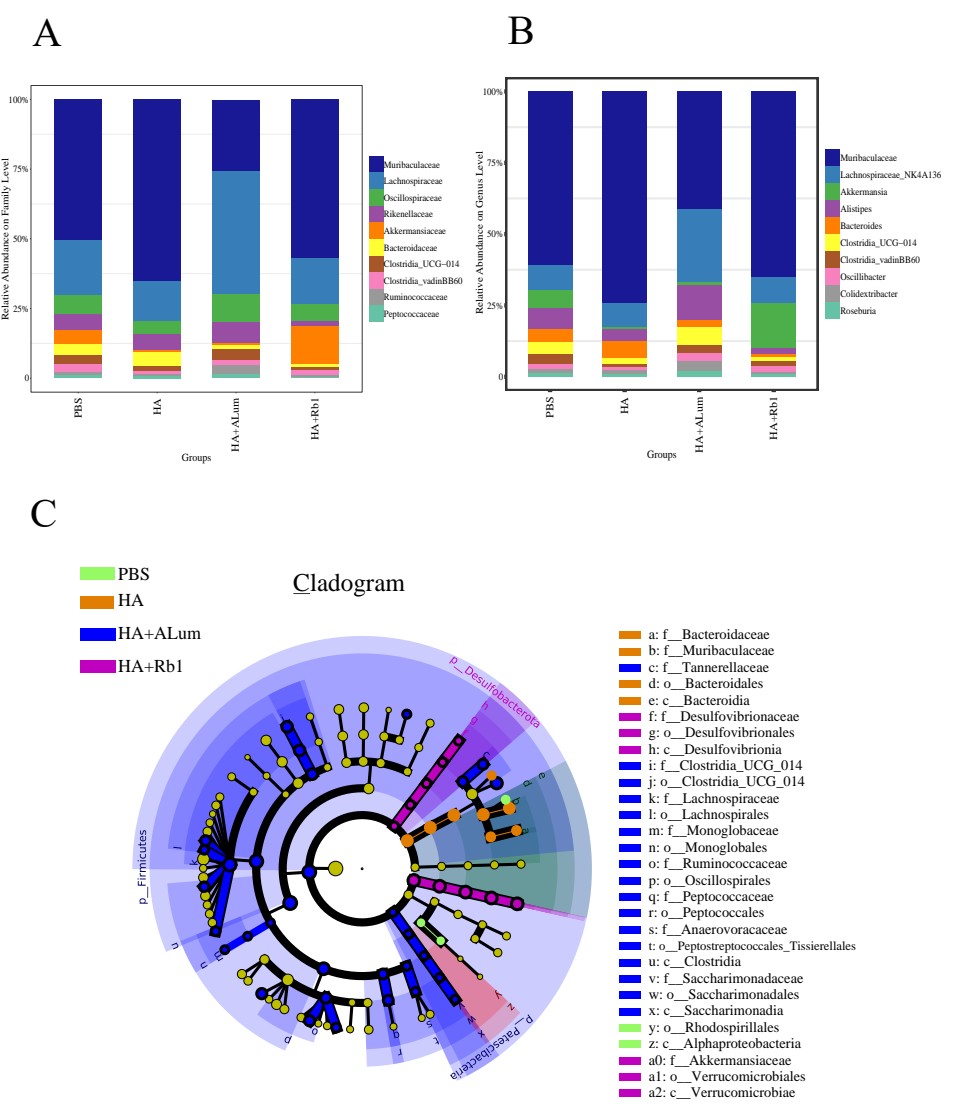

**Figure 5** **Ginsenoside Rb1 altered the dominant bacteria and microbiota composition in immunized mice.** The community abundance at the Family (A) and Genus (B) levels in control (PBS), the model (HA), the Alum administration (HA+Alum), and the Rb1 administration groups (HA+Rb1). (C) Species marker evolution trees for each group. The circles represent, from inside to outside, the taxonomic levels by phylum to genus (or species). Circle diameter size is proportional to relative abundance size. Species with no significant differences are colored in yellow, and the group colors those with differences.

Medicine (TCM) polysaccharides as adjuvants of vaccines to verify the efficacy of immune enhancement, and it is also a research hotspot of adjuvants in natural materials (*Kumar et al., 2022*; *Wan et al., 2022*).

Ginsenosides play an essential role in enhancing humoral and cellular immunity. *Rivera, Hu & Concha (2003)* showed that the antibody titers induced by Rb1 tested at a concentration of 830 μg per dose were significantly higher than those induced by Al (OH)$_3$ adjuvant. This finding is consistent with our experiment, which explains why there is no

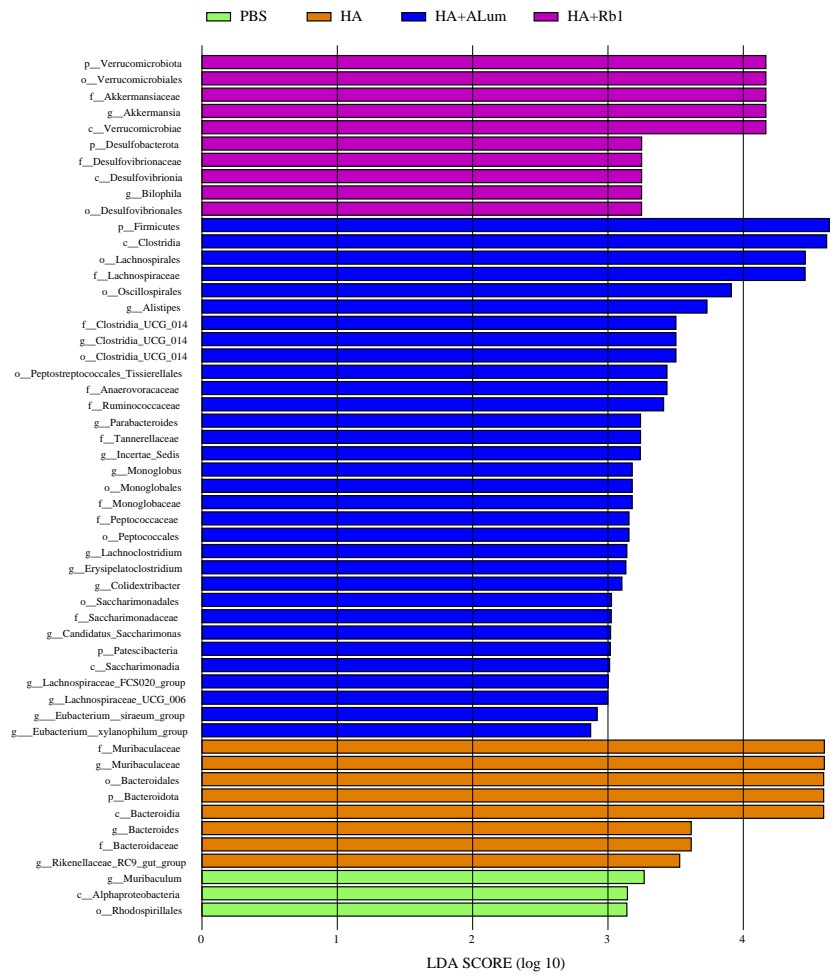

**Figure 6** **Biomarkers with significant abundance differences among groups (LAD value > 2).** Column lengths represent the contribution of biomarker species.

difference in antibody titers between the HA+Rb1-600 µg and HA+ALum groups. *Sun, Hu & Song (2007)* evaluated the adjuvant effect of Rb1 on the immune responses to ovalbumin (OVA) in mice, and this finding is consistent with our experiment, indicating that Rb1 can promote antibody production. *Rivera et al. (2005)* showed that ginsenoside Rb1 could elevate IL-4 and IL-10 levels of a swine parvovirus disease vaccine, and antibody titers were not decreased after five weeks. Cytokine elevation may strengthen the early immune response and produce more antibodies. However, the inflammatory factor expression level *in vivo* after infection was rarely studied, so we tested the serum inflammatory factor after H1N1 virus infection, and the result showed that RB1 reversed the high inflammatory factor level produced by the virus.

*Lu et al. (2020)* evaluated that Rb1 can reduce the expression levels of key inflammatory cytokines TNF-α and IL-6 in a cancer cachexia mouse model, this finding is consistent with our experiment, indicating that Rb1 can alleviate symptoms caused by inflammation.

*Kang et al (2021)* showed that ginsenoside Rb1 had antiviral activity in enterovirus-71 infected mice with a dose–effect relationship. Our study found that ginsenoside Rb1 had a dose–effect relationship regarding IgG antibody, rate of body weight change, lung tissue pathology, and IL-6 and TNF-α inflammatory cytokine expression. *Xin et al. (2019)* showed that ginsenoside Rb1 could enhance macrophage phagocytosis, and the mechanism may be related to the MAPK pathways. *Song et al. (2022)* showed that in an infectious bursal disease vaccine study using a Chitosan/Calcium Phosphate nanoparticle and combined ginsenoside Rb1 could induce the activation of chicken dendritic cells, upregulate the expression of MHC-II and CD80, and increase IL-1β and TNF-α levels, induce higher levels of specific antibody IgG and a higher IgG2a/IgG1 ratio, and promote long-term immune responses. Our study did not explore the mechanism, such as directly stimulating B lymphocyte immune response or enhancing antigen-presenting activity, stimulating helper T cells, and then enhancing humoral immune response, which may shed light on the immune elevating effect of ginsenoside Rb1.

Ginsenoside could regulate intestinal microbiota structure, and in previous studies (*Kang et al., 2016*; *Zheng et al., 2021*; *Zhu et al., 2021b*) it has been demonstrated that the efficacy of ginsenosides changes with the gut microbiota. In this study, we explored for the first time that ginsenoside Rb1 enhances the antibody enhancement of influenza vaccines in mice, as well as in the diversity and abundance of gut microbiota. Our study found that the abundance of *Akkermansiaceae* increased after the Rb1 intervention at the Family and Genus levels, and the abundance of *Murbaculaceae* in the HA+Alum group decreased compared to the HA group, but this can be reversed after Rb1 intervention. The dominant flora of the Rb1 group included *Desulfovibrionaceae*, *Desulfovibrionales*, *Desulfovibrionia*, *Akkermansiaceae*, Verrucomicrobiales, and Verrucomicrobiae. *Zhu et al. (2021a)* showed that ginsenosides could facilitate the anti-tumor efficacy of Cyclophosphamide in mammary carcinoma mice, and promote the reproduction of gut probiotics including *Akkermansia*, *Bifidobacterium*, and *Lactobacillus*, which is consistent with our experimental results on gut microbiota. *Yousuf et al. (2022)* showed that Rg1 enhances immunity by regulating gut microbiota homeostasis in mice, such as *Alistipes*, *Ruminococcaceae*, *Lachnospiraceae*, and *Roseburia* were increased, and the potential pathogens like *Helicobacteraceae*, *Dubosiella*, *Mycoplasma*, *Alloprevotella*, and *Allobaculum* were decreased. *Huang et al., (2022)* showed that ginseng polysaccharide improved the sensitivity of antitumor responses by modulating gut microbiota and increased the abundance of *Parabacteroides* and *Bacteroides*. *Zhou et al. (2021)* showed that ginseng polysaccharides could increase various beneficial mucosa-associated bacterial taxa *Clostridiales*, *Bifidobacterium*, and *Lachnospiraceae*, and decrease harmful bacteria *Escherichia-Shigella* and *Peptococcaceae*. *Sun et al. (2018b)* showed that ginseng extracts decreased the abundance of TM7 while *Proteobacteria*, *Methylobacteriaceae*, *Parasutterella*, and *Sutterella* increased, IgA levels were significantly elevated, which highly correlated with the abundance of *Bifidobacterium* and *Lactobacillus*.

The gut microbiota acts as mucosal adjuvants enhancing acquired immune responses and improving public health. *Hemmi et al. (2023)* showed that yogurt intake fermented with *L. bulgaricus* could reduce symptoms due to respiratory infections in a double-blind, randomized controlled trial. *Nagai, Moriyama & Ichinohe (2021)* showed that intranasal

supplementation of cultured oral bacteria from human volunteers could enhance antibody responses to the intranasally administered vaccine, such as the influenza A virus. *Hagan et al. (2019)* used antibiotics to reduce gut microbiota diversity and richness and then focused on H1N1-specific antibody responses in subjects with low antibody titers, demonstrated significant impairment and revealed a significant association between bacterial species and metabolic phenotype, highlighting the crucial role of the microbiome in modulating human immunity. *Akatsu et al. (2016)*, *Nagafuchi et al. (2015)* suggest that prebiotics could increase the intestinal microbiota of Bacteroides count in elderly individuals and then maintain the antibody titers of H1N1, H3N2, and B antigens.

Our study also had several limitations. Firstly, we should increase the experiment's sample size to meet serum requirements in multiple tests, which can increase the experimental content of antibody-dependent cell-mediated cytotoxicity (ADCC). In subsequent experiments, it is worth expanding from young mice to elderly mice to examine the trend of immunogenicity of Rb1 with age and verify the differences in the immune enhancement of Rb1 in mice of different age groups. Secondly, we did not design experiments on the long-term immunological effects of the vaccine, so in future experiments, the antibody duration can be increased to 6 months to observe the long-term effect of Rb1. Thirdlydetecting antibody IgG and typing IgG, we found that the difference in the Rb1 intervention group was difficult to detect as in previous studies (*Cheong et al., 2021*), which is related to the dosage of HA and the positive threshold we set. We believe that although the difference is not statistically significant, in future experimental designs, we can improve this situation by increasing the sample size and changing the dosage of Rb1 or HA. Moreover, adding content, such as dendritic cell maturation and effector cell detection, some inflammatory cytokines, and chemokines data in bronchoalveolar lavage fluid (BALF), could add further experimental confidence.

## CONCLUSION

In summary, this study proposes ginsenoside Rb1 as an adjuvant for the H1N1 split vaccine. Compared to the antigen group, it can boost IgG and antibody subtypes, improve the survival of mice after virus challenge, and reduce inflammatory cytokines expression. Furthermore, 16S rDNA sequencing showed that Rb1 reduced species diversity and increased species richness compared to the control group. Ginsenoside Rb1, as a biomaterial derived from TCM ginseng, has been disclosed for the first time as a novel adjuvant for the H1N1 influenza vaccine. In the future, it will be used to develop safe and effective vaccines and continue to explain its unique role in vaccines from the perspective of gut microbiota, which will also address the vaccine supply dilemma in pandemic outbreaks.

## ACKNOWLEDGEMENTS

We are grateful to Professor Haibo Wu for providing operational guidance

### Funding

This work was supported by the Science and Technology Project of Zhejiang Province (Grant no. 2023ZL047), and the Key Research Projects of Zhejiang Chinese Medical University (Grant no. 2022FSYYZZ03). The funders had no role in study design, data collection and analysis, decision to publish, or preparation of the manuscript.

### Grant Disclosures

The following grant information was disclosed by the authors:
The Science and Technology Project of Zhejiang Province: 2023ZL047.
Key Research Projects of Zhejiang Chinese Medical University: 2022FSYYZZ03.

### Competing Interests

The authors declare there are no competing interests.

### Author Contributions

- Chuanqi Wan conceived and designed the experiments, performed the experiments, prepared figures and/or tables, authored or reviewed drafts of the article, and approved the final draft.
- Rufeng Lu performed the experiments, prepared figures and/or tables, and approved the final draft.
- Chen Zhu conceived and designed the experiments, prepared figures and/or tables, and approved the final draft.
- Haibo Wu conceived and designed the experiments, performed the experiments, authored or reviewed drafts of the article, and approved the final draft.
- Guannan Shen analyzed the data, prepared figures and/or tables, and approved the final draft.
- Yang Yang analyzed the data, prepared figures and/or tables, and approved the final draft.
- Xiaowei Wu analyzed the data, prepared figures and/or tables, and approved the final draft.
- Bangjiang Fang conceived and designed the experiments, authored or reviewed drafts of the article, and approved the final draft.
- Yuzhou He conceived and designed the experiments, authored or reviewed drafts of the article, and approved the final draft.

### Animal Ethics

The following information was supplied relating to ethical approvals (i.e., approving body and any reference numbers):

The Animal Ethics Committee of Zhejiang Chinese Medical University reviewed and approved the animal study (IACUC-20220725-37).

## DNA Deposition

The following information was supplied regarding the deposition of DNA sequences:

The sequencing data of mouse gut microbiota is available at SRA: SRP442547. https://www.ncbi.nlm.nih.gov/sra/PRJNA979777.

## Data Availability

The raw measurements are available in the Supplementary File.

## Supplemental Information

Supplemental information for this article can be found online at http://dx.doi.org/10.7717/peerj.16226#supplemental-information.

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
