# Peer review of "Ginsenoside Rb1 enhanced immunity and altered the gut microflora in mice immunized by H1N1 influenza vaccine"

_PeerJ, doi:10.7717/peerj.16226_

## Round 0.1 · original submission · Major Revisions

Dear Dr. He

Thank you for submitting your manuscript to PeerJ. While the reviewers found the paper of interest, there are major revisions required. Please pay special attention to the comments and, when submitting your revised manuscript, be sure to include a letter detailing your response to each concern Thank you for submitting your manuscript to PeerJ and we look forward to receiving your revised manuscript.

·

Basic reporting

The article demonstrates a high level of professionalism in its use of English, incorporation of relevant literature references, adherence to journal criteria for structure, and support of results with data.

However, there are a few areas where the article can be improved:
1) Clear statement of the research objectives and hypothesis would provide readers with a better understanding of the study's purpose and the specific questions being addressed.

2) Recommend to carefully review the article for any typographical errors or language issues. For example, there are typos in lines 106, 112, and 113 as well as Table 1 that should be corrected to enhance the overall quality of the manuscript

3) Certain sections of the article contain technical language that may be difficult for some readers to understand. To improve accessibility, it would be helpful to provide the full form of acronyms wherever possible, especially when they are first introduced in the paper. For instance, "ASVs", "SEM" and "TCM" should be written as "Amplicon Sequence Variant", 'Standard Error of mean" and "Traditional Chinese Medicine," respectively.

4) Figure 4A-4D, does not specific what dose group of HA-Rb1 is being used.

Experimental design

The experimental design is thoroughly described, encompassing the animal experiments, reagents, data collection methods, and data analysis techniques.

However, there are a few areas that could be improved.

1) Provide a clear and concise description (the language can be improved) of the experimental design, specifically explaining how the animals were divided into different groups and tests. This clarification will help readers understand the allocation and grouping of animals, ensuring transparency in the experimental process.

2) To enhance the reproducibility of the study, it is recommended to include specific details about the brands and sources of the chemicals and antibodies used. Providing this information will allow other researchers to precisely replicate the experimental conditions and ensure consistency across studies.

Validity of the findings

The article emphasizes the importance of adjuvant studies to enhance the immune response to viral infections where the authors mainly highlight the potential of polysaccharides extracted from TCM plants such as ginsenoside Rb1, as adjuvants for vaccines.

Some comments:
1) The results suggest that even aluminum as a adjuvant provides same antibody response at a lower dose (100ug) compared groups with Rb1. Can you add some more details on why we think Rb1 can be more beneficial? Why chose Rb1 at a higher dose when you get the same antibody levels with Aluminum at a lower dose?

2) Some of the statements needs some context to support their conclusions - For eg. line 160 it says high mortality rate (28.8%) in the Alum group highlight the predominance of mice in RB1 group - can you provide reference to this conclusion?

3) While the discussion references previous studies, please provide more context and integrate these studies more effectively. For example, when discussing the immune-enhancing effects of ginsenoside Rb1, the authors could compare and contrast their findings with previous studies to highlight the novelty or consistency of their results.

4) Authors briefly mentions about limitations of the study, but does not provide thorough analysis of their impact - how these limitations may have influenced their results and interpretations.

5) The conclusion section is missing the excerpt provided. It would be important to include a concise summary of the main findings and their implications. The conclusion should also reiterate the significance of the study's contributions to the field and potentially suggest practical applications or further research directions.

Reviewer 2 ·

Basic reporting

In this manuscript, authors explored the role of Ginsenoside Rb1 and claimed Ginsenoside Rb1 boosted IgG and antibody subtypes in the H1N1 infuenza vaccine. They further show that it improved survival of mice after virus challenge and reduced inflammatory cytokines expression. Furthermore, they also show by 16SrDNA Rb1 decreased species diversity, but increased species richness compared to the control group. However, second part of results where they show that Ginsenoside Rb1 alters the gut microflora in mice has already been shown before which makes it not novel. These studies (PMID: 36238549, PMID: 34899310) have shown that Ginsenoside Rb1 changes the composition and function of gut microbiota and function which makes the results of altered gut microflora not novel. Authors should do significant number of experiments to prove that Ginsenoside Rb1 enhanced immunity and show more experimental data from the lung.

Experimental design

Introduction: Authors should re-write the introduction in more elaborate way. I suggest that you also cite the papers that have shown that Ginsenoside Rb1 changes the microbial composition.

Results:

Figure 1, Color of the bar in the graphs is not consistent. For example, blue bar (PBS) in Fig.1b is blue whereas in other graphs HA group is blue. It is confusing to understand. Please keep the color of the bar or individual data points from 1B-1G same.

Figure 2, Data shown in figure 2 is not sufficient to claim that Ginsenoside Rb1 attenuates severe lung injury during infuenza virus infection.
• Please show inflammatory cytokine and chemokines data in bronchoalveolar lavage fluid (BALF)
• To claim that GinsenosideRb1 enhanced immunity authors need to show data of immune cells such as neutrophils, alveolar macrophages or other immune cells.
• Please include the labels of H and E staining from A-F.

Validity of the findings

No comments

---

## Round 0.2 · accepted · Accept

Thank you for addressing the concerns of the reviewers. I am happy to inform you that your manuscript is now acceptable for publication.

·

Basic reporting

No comments

Experimental design

No comments

Validity of the findings

No comments

Additional comments

Dear Authors,

I hope this message finds you well. I had the opportunity to review the revisions you have made to your manuscript based on my previous suggestions and comments. I must express my sincere appreciation for the effort and thoroughness with which you have addressed each point.

The additional data and information you've incorporated have greatly enhanced the depth and clarity of the paper. Furthermore, your detailed and thoughtful responses to my comments have provided convincing answers and have successfully resolved the concerns I had.

Given the substantial enhancements and the significant contribution this research makes to our field, I am pleased to recommend that the manuscript be accepted for publication.

Thank you once again for your diligence in addressing the revisions. I look forward to seeing your valuable work published and am confident it will be well-received by the scientific community. Good luck.